# *Arabidopsis thaliana SHOOT MERISTEMLESS* Substitutes for *Medicago truncatula SINGLE LEAFLET1* to Form Complex Leaves and Petals

**DOI:** 10.3390/ijms232214114

**Published:** 2022-11-15

**Authors:** Véronique Pautot, Ana Berbel, Thibaud Cayla, Alexis Eschstruth, Bernard Adroher, Pascal Ratet, Francisco Madueño, Patrick Laufs

**Affiliations:** 1Université Paris-Saclay, INRAE, AgroParisTech, Institut Jean-Pierre Bourgin (IJPB), 78000 Versailles, France; 2Instituto de Biología Molecular y Celular de Plantas, Consejo Superior de Investigaciones Científicas-Universidad Politécnica de Valencia, Campus de la Universidad Politécnica de Valencia, 46022 Valencia, Spain; 3Université Paris-Saclay, CNRS, INRAE, Université Evry, Institute of Plant Sciences Paris-Saclay (IPS2), 91405 Orsay, France; 4Université de Paris, Institute of Plant Sciences Paris-Saclay (IPS2), 91405 Orsay, France

**Keywords:** *Medicago truncatula*, meristematic activity, flower development, SINGLE LEAFLET/LEAFY, SHOOT MERISTEMLESS transcription factors

## Abstract

LEAFY plant-specific transcription factors, which are key regulators of flower meristem identity and floral patterning, also contribute to meristem activity. Notably, in some legumes, *LFY* orthologs such as *Medicago truncatula SINGLE LEAFLET* (*SGL1*) are essential in maintaining an undifferentiated and proliferating fate required for leaflet formation. This function contrasts with most other species, in which leaf dissection depends on the reactivation of *KNOTTED*-*like* class I homeobox genes (*KNOXI*). *KNOXI* and *SGL1* genes appear to induce leaf complexity through conserved downstream genes such as the meristematic and boundary *CUP-SHAPED COTYLEDON* genes. Here, we compare in *M. truncatula* the function of *SGL1* with that of the *Arabidopsis thaliana KNOXI* gene, *SHOOT MERISTEMLESS* (*AtSTM)*. Our data show that *AtSTM* can substitute for *SGL1* to form complex leaves when ectopically expressed in *M. truncatula*. The shared function between *AtSTM* and *SGL1* extended to the major contribution of *SGL1* during floral development as ectopic *AtSTM* expression could promote floral organ identity gene expression in *sgl1* flowers and restore sepal shape and petal formation. Together, our work reveals a function for *AtSTM* in floral organ identity and a higher level of interchangeability between meristematic and floral identity functions for the AtSTM and SGL1 transcription factors than previously thought.

## 1. Introduction

Meristems are essential for plant development, as they are required for the continuous growth and development that are distinguishing features of plants. Amongst all the different types of meristems, the shoot apical meristem (SAM) and the floral meristem (FM) share many features and have been well characterized. The class I *KNOTTED*-like homeobox (*KNOX1*) *SHOOT MERISTEMLESS* (*STM*) and *CUP*-*SHAPED COTYLEDON* (*CUC1), CUC2* and *CUC3* genes are essential regulators of meristem and boundary activities in *Arabidopsis thaliana* (*A. thaliana*) [1,2]. Boundaries are domains of restricted growth located between the meristem and initiating organ primordia or between two organs. These domains control organ separation, inflorescence architecture, organ abscission, fruit opening and leaf shape. Boundaries share overlapping features with meristems, and the regulation of both involves common factors [3]. *CUC* genes are required for SAM initiation and establish boundaries together with *STM*, which is in turn required for SAM maintenance [4,5,6,7,8,9,10,11] The three *A. thaliana CUC* genes, *CUC1*, *CUC2* and *CUC3*, share partially redundant roles, while also having specific functions. *CUC1* and *CUC2* but not *CUC3* transcripts are negatively regulated by *microRNA164* (*miR164*) [12,13].

LEAFY (LFY) is a key regulator of flower meristem identity and floral patterning [14,15,16,17]. LFY acts as a pioneer transcription factor and promotes chromatin accessibility to its target genes *APETALA1* (*AP1*) and *AGAMOUS* (*AG*) [18,19]. LFY also contributes to meristem function, particularly to the formation of floral meristems in *A. thaliana* [20,21,22,23]. LFY acts together with UNUSUAL FLOWER (UFO), an F-box protein, which is a substrate adaptor of CULLIN1–RING ubiquitin ligase complexes (CRL1) [24,25] to control meristem function and identity [17,26]. In flowers, both LFY and auxin transport contribute to proper positioning of sepal primordia through the regulation of *CUC2* expression [27]. Besides determining floral identity and patterning, LFY also contributes to the meristematic identity of floral or axillary meristems with several regulators [21,28,29,30,31]. Among them are PENNYWISE (PNY) and POUNDFOOLISH (PNF), two BEL1-like (BELL) homeodomain partners forming heterodimers with STM [32,33]. The LFY implication in axillary meristem emergence is mediated through REGULATOR OF AXILLARY MERISTEMS1 (RAX1), a MYB transcription factor [34,35], and through the repression of *ARABIDOPSIS RESPONSE REGULATOR* (*ARR7*), encoding a cytokinin signaling component [21,36].

Meristematic features can also be found outside *bone fide* meristems, such as in the leaves. This attribute is particularly obvious in compound leaves, in which leaflet formation requires a transient maintenance of a meristematic-like stage. Indeed, in most species with compound leaves, *KNOX1* gene down-regulation at leaf initiation is only transient and these genes are reactivated following leaf initiation, leading to leaflet formation [37,38,39,40]. In the inverted repeat-lacking (IRLC) clade of legume species, the formation of compound leaves requires the activity of the *LFY* orthologs called *UNIFOLIATA* (*UNI*) in pea (*P. sativum*) and *SINGLE LEAFLET1* (*SGL1*) in *Medicago truncatula* (*M. truncatula*) [41,42]. These *LFY* orthologs substitute for *KNOX1* expression, which is permanently excluded from the initiating leaf primordia [43,44]. *SGL1* is expressed in the entire SAM and highly expressed in developing leaves, where its prolonged expression is required for the formation of compound leaves [42,45]. However, ectopic expression of *KNOX1* genes in *M. sativa* and *M. truncatula* leaves can further increase leaf dissection [44,46], suggesting that these two *Medicago* species retain the capacity to respond to both *LFY* and *KNOX1* pathways. Consistent with a central role of *LFY* orthologs in IRLC legume leaf morphology, loss of function of the pea *UFO* ortholog, *STAMINA PISTILLOIDA* (*STP*), leads to leaf complexity reduction [47]. In contrast, in non-IRLC legumes such *Lotus japonicus* and soybean, *LFY* orthologs only play a minor role, and KNOX1 proteins accumulate in leaves and are likely associated with compound leaf development [44,48].

In simple leaves, such as in *A. thaliana*, repression of *KNOX1* genes is permanent, limiting their meristematic features [49,50]. However, these leaves are still able to develop an increased complexity in response to ectopic expression of *KNOX1* genes [40,51,52,53] and to *UFO* [54].

The observation that depending on the species, *LFY* and *KNOX1* genes can similarly increase leaf complexity (through the formation of leaflets or serrations) and that some species are able to respond to both factors, suggests that both pathways may at least partially converge to control leaf development. *CUC* genes could be such a convergence point as both *KNOX1* and *LFY* pathways require the activity of CUC2 to make compound leaves [55,56]. Similar to *CUC1/2* in *A. thaliana*, the expression of the *M. truncatula NO APICAL MERISTEM* (*MtNAM*) ortholog is regulated by *miR164* [57], and *MtNAM* is required to maintain boundaries both for cotyledon and leaflet separation besides its role in apical meristem initiation [58]. SGL1 function in leaflet primordium initiation is epistatic to MtNAM activity and *MtNAM* RNAs levels are reduced in *sgl1* mutant [58], suggesting that *SGL1* acts upstream of *MtNAM* in this species.

Besides its function in leaves, SGL1 also plays a role in floral meristem identity. *M. truncatula* is a legume species developing compound inflorescences. Upon floral transition, the shoot apical meristem transforms into a primary inflorescence meristem (I1) and gives rise to a lateral secondary inflorescence meristem (I2), which produces a bract, one to three flowers and a spike [59,60,61]. In contrast to other flowering species which show a sequential floral ontogeny with successive formation of sepals, petals, stamens and carpels, each floral organ derives from a specific primordium; petals and sepals differentiate from common primordia in *M. truncatula* [59]. Thus, each floral meristem gives rise sequentially to five sepals and four common primordia, which further differentiate into five petals and ten stamens, and one carpel. The *Arabidopsis* floral organ identity genes are conserved in legumes [62]. Loss of function of *SGL1* leads to the reversion of common primordia into incomplete floral meristems, giving rise to sepals and carpels without petals and stamens [42]. This phenotype is related to a B function loss. Similar to *LFY* in *Arabidopsis* [63], *SGL1* acts synergistically with *MtPROLIFERATING INFLORESCENCE MERISTEM (MtPIM),* the *A. thaliana* ortholog of *APETALA1 (AP1)* in *M. truncatula* to determine floral meristem identity [61,64]. *MtNMH7* and *MtTM6* are the *A. thaliana AP3-like* paralogs. *MtNMH7* determines petal identity whereas *MtTL6* controls stamen identity [65]. *MtPISTILLATA* (*MtPI*) and *MtNGL9* are the two *A. thaliana PI*-like paralogs, with *MtPI* functioning as the master regulator of B function [66,67]. The *M. truncatula* genome harbors two redundant *MtAG* members, *MtAGa* and *MtAGb*, which specify stamen and carpel identity and floral meristem determinacy [68,69]. Recently, a novel regulator of inflorescence development and floral organ identity was identified in *M. truncatula:* the *AGAMOUS-like FLOWERS* (*AGLF*) gene, which encodes a MYB domain protein that promotes the C floral identity function besides repressing A and B functions [69,70].

Here, we further compared the meristematic activity of *SGL1* (*LFY*) and *AtSTM* (*KNOX1*) using *M. truncatula* compound leaf as a model system. We first showed that *AtSTM* could substitute for *SGL1* to form complex leaves. We next tested whether *AtSTM* could also substitute for *SGL1*’s role during floral development. Indeed, *AtSTM* expression could restore petal formation in *sgl1* flowers, revealing that AtSTM could substitute for SGL1 function to specify petal identity and promote floral organ identity gene expression. Therefore, our data reveal a high level of interchangeability between SGL1 and KNOX1 activities in *M. truncatula* that extends beyond the generally accepted meristematic function to the determination of the identity and growth of the flower perianth.

## 2. Results

### 2.1. AtSTM Substitutes for SGL1 in M. truncatula to Form Compound Leaves

The *M. truncatula* genome harbors two *MtSTM-like* genes, *MtKNOX1* and *MtKNOX6,* and a previous report describes in vitro plantlets overexpressing the *MtSTM-like* genes, *MtKNOX1* and *MtKNOX6*, in *M. truncatula* [46]. However, only the vegetative phenotype was described, as the phenotype of *MtKNOX1* and *MtKNOX6* overexpressors was extremely severe. Therefore, to overcome such strong phenotypes, we thought to use a *KNOX* gene from a heterologous system. *AtSTM* shares 62.8% amino acid identity with *MtKNOX1* and 64.86% with *MtKNOX6*, and in addition to modifying leaf shape when ectopically expressed, *AtSTM* also has an established role in *Arabidopsis* floral identity [5,6,32,52,71,72]. Thus, we selected *AtSTM* to be expressed in *M. truncatula* and to explore its potential more widely; we expressed it under two different promoters by generating the *p35S:AtSTM* and *pSGL1:AtSTM* constructs that we first introduced in wild-type plants (see Section 4 and Appendix A).

Transgenic lines expressing high levels of *AtSTM* presented a severe phenotype and were not viable in the greenhouse, similar to in vitro plantlets overexpressing *Mt-KNOX1*-like genes [46] (Appendix A). Only transgenic plants with low levels of *AtSTM* expression could be investigated (Figure 1 and Appendix A). The overall development of these lines was quite normal, although their fertility was reduced. In wild-type *M. truncatula*, the juvenile first leaf is simple, while adult later leaves are trifoliate and composed of a terminal leaflet with two lateral leaflets (Figure 1A–D). Ectopic *AtSTM* under the *p35S* or the *pSGL1* promoters occasionally led to the formation of an additional leaflet fused to the terminal leaflet of adult leaves (Figure 1F,H and Appendix A). Quantitative analyses were performed using the *p35S:AtSTM* line (Figure 1Q). The wild-type first leaves (rank 1) were simple, while the majority of adult leaves (ranks 2 to 5) were trifoliate (only 4 out of 72 leaves had more than three leaflets). The *p35S:AtSTM* sequences seldom led to complex leaves, as only 8 out of 72 adult leaves (ranks 2 to 5) were more complex (Figure 1Q).

We then tested whether *AtSTM* expression is sufficient to rescue the *sgl1* leaf phenotype (see Section 4). In the *sgl1* mutant, the majority of leaves are simple (Figure 1I–L). All rank 5 leaves were simple, but 12 out 54 leaves (ranks 2 to 4) were bi- or trifoliate in the *sgl1* mutant (Figure 1Q). In contrast, in *p35S:AtSTM sgl1* plants, the majority of adult leaves were trifoliate as in wild-type (Figure 1B–D,O,P). The *p35S:AtSTM* construct restored almost systematically the capacity to form trifoliate leaves, with 52 out of 54 leaves (ranks 3 to 5) producing at least three leaflets (Figure 1Q). Therefore, we concluded that *AtSTM* can replace *SGL1* to promote leaflet formation.

To explore the developmental origin of the extra or rescued leaflets in the different backgrounds, we imaged by SEM young developing leaf primordia (Figure 2). As observed in wild-type apices, a pair of lateral leaflets and a terminal leaflet initiated in *AtSTM* transgenic lines during early leaf primordium development (Figure 2A,C). At stage S8, additional leaflets could form at the base of the terminal leaflet in *AtSTM* (arrows Figure 2D), which were not observed in the wild-type (Figure 2B) and therefore resulted from secondary morphogenesis. This indicates that the morphogenetic window during which leaflets can be initiated is extended following *AtSTM* expression. In *p35S:AtSTM sgl1* plants, the terminal primordium was surrounded by two lateral primordia (Figure 2G,H), already visible at early stages (S4), as seen in the wild-type (Figure 2A,G). Thus, leaflet restoration in *p35S:AtSTM sgl1* does not appear to rely on a late production of leaflets but a rescue of the normal developmental process with a restoration of early lateral leaflet initiation, as occurs in the wild-type.

### 2.2. AtSTM Substitutes for SGL1 in M. truncatula in Specifying Petal Formation

*M. truncatula* is a legume species developing compound inflorescences. The wild-type *M. truncatula* mature flower (Figure 3A–I) comprises four whorls consisting of a calyx formed by five sepals fused at their base (Figure 3B), a corolla containing three types of yellow petals, the standard or the vexillum at the adaxial position (Figure 3C), the keel formed by two fused petals at the abaxial position surrounded by two lateral petals and the alae or wings (Figure 3D–G). The third whorl consists of an independent stamen filament at the adaxial position, the vexillary stamen filament and nine stamen filaments fused into a staminal tube that surrounds a monocarpous gynoecium [59] (Figure 3H,I). The *sgl1* mutants produce inflorescences with cauliflower-like floral structures, containing incomplete floral meristems (FMs), elongated sepals and occasionally carpels [42] (Figure 3U). These cauliflowers do not produce petals nor stamens, similar to *lfy* mutants in *Arabidopsis* [73].

Wild-type plants for *SGL1* overexpressing *AtSTM* occasionally produced abnormal flowers showing fused organs and are characterized by an increase in petal identity with petaloid sepals and petaloid stamens (Figure 3K–S). Some petals showed alterations in shape or serrated margins (Figure 3L,M). These flowers occasionally produced two to three unfused carpels (Figure 3M–O). Flowers can show petaloid sepals (Figure 3N,S), petaloid stamens (Figure 3O,P) and petaloid carpels (Figure 3Q). The fertility was severely reduced, with some plants infertile. The fruits were small, with fewer discs and unbent spines compared with wild-type fruits (Figure 3T,J). These fruits contained a few seeds. The same phenotypes were occasionally observed in *pSGL1:AtSTM* flowers (Appendix A). We then tested the effects of *p35S:AtSTM* on *sgl1* flower development. Surprisingly, the ectopic expression of *AtSTM* rescued sepal shape and petal formation in the *sgl1* mutant (Figure 3V–Y). Similar to wild-type flowers, *p35S:AtSTM sgl1* flowers formed a calyx with five sepals fused at their base (Figure 3B,W). Inside the calyx, the *p35S:AtSTM sgl1* flowers showed a cauliflower phenotype with incomplete FMs, producing a few sepals and a majority of petals or petals with sepal sectors. Petals were partially restored as some of them had a vexillium-like, wing-like or keel-like shape (Figure 3X,Y). Thus, when ectopically expressed, *AtSTM* restores petal formation in *sgl1*. These flowers did not form carpels, in contrast to *sgl1* flowers, suggesting a deficiency in C function (Figure 3). The majority of organs formed were petals, as one cauliflower flower from a *35S:AtSTM sgl1* line could produce up to 65 petals (Appendix A).

SEM analyses were performed to further characterize these flowers at early developmental stages. Figure 4A–D shows wild-type floral development. At stage 4, the wild-type floral meristem had formed five sepal primordia, four common primordia and a carpel primordium (Figure 4B). At late stage 5, the wild-type floral meristem displayed the complete set of floral organ primordia, with petal and stamen primordia deriving from the differentiation of common primordia (Figure 4D). Figure 4E–H shows the floral development of a *p35S:AtSTM* plant wild-type for *SGL1*. Figure 4F shows a late stage 5 *p35S:AtSTM* floral meristem. Based on sepal development, a delay in the formation of the inner floral organ primordia could be observed compared with the wild-type (Figure 4F,D). In contrast, Figure 4G shows a stage 5 floral meristem containing differentiated petals and stamen primordia and two carpel primordia, indicating that the delay in internal organ primordia differentiation is variable between flowers. Figure 4H shows a *p35S:AtSTM* flower developing two carpels. Similar to previous data [42,61], *sgl1* inflorescences showed multiple incomplete FMs, elongated sepals, defective common primordia and carpel primordia (Figure 4I–L). Sepal primordia further develop into elongated sepals and carpel primordia into a carpel-like structure. The cauliflower phenotype is caused by the iterative conversion of common primordia into incomplete floral meristems (Figure 4I,L). In *sgl1* mutants overexpressing *AtSTM* (Figure 4M–O), the sepal form was restored, suggesting that *AtSTM* could take over SGL1 function for the control of sepal shape. The late stage 5 floral meristems showed a delay in the differentiation of other floral organ primordia, as observed in *p35S*:*AtSTM SGL1* plants (Figure 4N,F). Later, petals and sepals differentiated from these primordia (Figure 4O). Together, these observations show that expression of *AtSTM* partly restored normal early morphogenesis of *sgl1* flowers.

### 2.3. AtSTM Substitutes for SGL1 to Promote Floral Organ Identity Gene Expression

To determine if AtSTM activates A and B functions to promote petal formation in *sgl1* flowers, we used in situ hybridization to analyze the expression pattern of floral organ identity genes in *p35S:AtSTM sgl1* flowers. We first investigated the expression of the A class gene *MtPIM,* the *A. thaliana* ortholog of *AP1* in *M. truncatula*. *MtAP1* has a conserved role with orthologous genes and is required to specify floral meristem and floral organ identity [61,64]. In wild-type inflorescences, *MtAP1* transcripts localize to the floral meristem and bract (Figure 5A,B). In a stage 4 flower meristem, *MtAP1* expression was observed in sepal primordia and was restricted to the outer domain of the common primordia that further gives rise to sepals and petals and was absent from the inner part, which differentiates into stamens and carpel (Figure 5C) [61,64]. At later stages, *MtAP1* expression was maintained in sepals and petals (Figure 5D). Similar to the pattern described in [61], in *sgl1* flowers, *MtAP1* was expressed in the floral meristem and in the bract (Figure 5E). *MtAP1* was expressed uniformly in defective common primordia and in reiterated floral meristems (Figure 5F–H). At later stages, *MtAP1* expression localized to the outer incomplete floral meristem and disappeared from the central domain that further differentiates into carpels (Figure 5F,G). In *p35S:AtSTM sgl1* flowers, *MtAP1* was more widely expressed than in *sgl1* flowers, with *MtAP1* detected in reiterated floral meristems and in developing petals (I–K). Thus, in *p35S:AtSTM sgl1* flowers, AtSTM acts as a positive regulator of A function, contributing to enhanced petal identity.

We then investigated the expression of the B class gene *MtPI*. In wild-type, *MtPI* transcripts were localized to common primordia cells and later restricted to petal and stamens (Figure 6A,B) and [66,67]. In the *sgl1* mutant, no *MtPI* expression was detected in defective common primordia, consistent with the phenotype of *sgl1* flowers, which lack petals and stamens (Figure 6C). In *sgl1* flowers overexpressing *AtSTM, MtPI* expression was detected in defective common primordia (Figure 6D,E inset-a). At a later stage, *MtPI* localized to the outer domain of the defective common primordia that further gives rise to petal-like organs (Figure 6E and inset-b). Later, *MtPI* is expressed in petal-like organs (Figure 6D,E). Thus, AtSTM acts as a positive regulator of *MtPI* expression consistent with the restoration of petal identity.

We further determined the expression of the *M. truncatula* ortholog of the *A. thaliana* C-class gene *AG*. *MtAGb* was used as a probe as its signal is stronger and it is more restricted than that of *MtAGa* [68]. In wild-type flowers, *MtAGb* expression was first detected at stage 2 in the central part of the floral meristem where the carpel will develop (Figure 7A). At stage 4, *MtAGb* expression was mainly localized to the inner domain of the common primordia that will further give rise to stamens and to carpel primordia (Figure 7B). At later stages, its expression was restricted to stamens, carpel and ovules (Figure 7C,D). In *sgl1* flowers, a weak signal was detected in floral meristems and defective common primordia and was absent in the L1 layer (Figure 7E–G). Later, its expression was detected in carpel-like structures and ovules (Figure 7H). In *sgl1* plants overexpressing *AtSTM, MtAGb* expression was detectable in only a few flowers (3 of 13). In these flowers, the signal was weak and restricted to a few cells in FM beneath the two outer most layers (Figure 7I).

## 3. Discussion

Here, we compared the activity of two transcription factors, AtSTM and SGL1, in *M. truncatula*. Our analysis is based on transgenic plants that were able to grow in a greenhouse and therefore expressed *AtSTM* at low levels. This allowed us to investigate the activity of *AtSTM* during flower development.

An increase in the leaflet number was only occasionally observed following *AtSTM* ectopic expression in wild-type *M. truncatula*. This limited effect of *AtSTM* could be linked to *AtSTM* expression levels in these lines, which were low. The additional leaflets were formed at the base of the terminal leaflet and resulted from a secondary morphogenesis. This suggests that AtSTM leads to additional leaflets through the extension of the meristematic activity, allowing more leaflets to emerge, and not from the division of the lateral leaflets into two structures. In *M. truncatula*, the terminal leaflet derives from the terminal zone where auxin maxima are located through the activity of *SMOOTH LEAF MARGIN1 (SLM1)*, the *PIN1* ortholog in *M. truncatula* [74]. Lateral leaflets result from the marginal blastozone activity and the formation of local auxin maxima that depend on SGL1 activity [74]. The tetrafoliate pattern seen in *AtSTM* transgenic lines likely results from a defect in auxin distribution in the terminal zone. This leaf patterning is also found in *M. truncatula* plants inactivated for *HEADLESS* (*HDL*) or *MtREVOLUTA1* (*MtREV1*), the putative orthologs of *A. thaliana WUSCHEL* and *REVOLUTA*, of which mutants are altered in auxin homeostasis [75,76]. The ectopic expression of *AtSTM* could rescue the formation of lateral leaflets in the *sgl1* mutant. These data show that *AtSTM* could substitute for *SGL1* via an independent pathway to form complex leaves. This suggests that AtSTM could bypass the requirement for SGL1 during the formation of compound leaves in *M. truncatula*, indicating shared functions between these proteins, a conclusion further reinforced by the study of the floral phenotype of *p35S:AtSTM sgl1* plants.

Our data revealed an unexpected effect of AtSTM on floral development, as AtSTM could induce petal identity. The effect of AtSTM on petal identity was moderately visible in an *SGL1* wild-type background, as only few chimeric petaloid floral organs were formed, but was dramatic in an *sgl1* mutant background. Indeed, all *p35S:AtSTM sgl1* flowers produced petals or petals with sepal sectors, while such organs were missing in *sgl1*. Although the increase in petal number could be in part due to the indeterminate state conferred by the *sgl1* mutation, it nevertheless indicates that AtSTM can restore petal formation in an *sgl1* mutant. The shape of *sgl1* sepals was also restored following *AtSTM* expression, showing that *AtSTM* could substitute for other functions of *SGL1* during flower development. The formation of petals in *p35S*:*AtSTM sgl1* was correlated with an activation of *MtAP1* and more notably of *MtPI* expression, suggesting that AtSTM could promote the expression of these floral organ identity genes to restore petals, and not through an indirect effect on floral meristem growth, for instance. Such a role for *KNOX1* genes in the promotion of B function was not yet reported in either *M. truncatula* nor in *A. thaliana* [44,46,72].

On the contrary, *p35S*:*AtSTM sgl1* flowers did not form stamens, and in contrast to *sgl1* cauliflowers, which developed carpels, *AtSTM sgl1* cauliflowers lacked carpels. *MtAGb* expression was only rarely detected in *p35S:AtSTM sgl1* cauliflowers, in agreement with the lack of carpel identity. Interestingly, the expression of *MtAGb* was systematically detected in floral meristems beneath the outermost layers in *sgl1* background. The localization and the low intensity of the *MtAGb* signal in *sgl1* cauliflowers suggest that SGL1 influences *MtAGb* expression.

In *Arabidopsis*, a link for *AtSTM* with carpel identity was revealed with the analysis of plants compromised for AtSTM activity in line with *AtSTM* expression in flowers [5,6,71,73]. A more direct contribution to carpel identity was illustrated with the phenotype of *A. thaliana KNOX1* overexpressors showing homeotic conversion of ovules into pistils. However, *KNOX1* ectopic expression does not complement the *ag* mutant [52,72]. In line with these conclusions made in *Arabidopsis*, we observed that in *M. truncatula*, ectopic expression of *AtSTM* could not induce the C function in the absence of SGL1 activity. It is possible that in *p35S:AtSTM sgl1* flowers, AG is playing a role related to floral meristem termination more than a function related to the specification of carpel identity.

The impact of *AtSTM* was more obvious both in leaves and flowers of the *sgl1* mutant compared with wild-type *SGL1* plants. This distinct impact could suggest that the STM pathway is more effective in the absence of SGL1 activity. It is likely that SGL1 acts in part through the *M. truncatula UFO* ortholog, as it does in *Arabidopsis* and other legumes. Indeed, in pea and in *Lotus japonicus* defective in *STAMINA PISTILLOIDA* (*STP*) or in *PROLIFERATING FLOWER ORGAN* (*PFO*), the *A. thaliana UFO* orthologs lack petals and stamens and show a reduced carpel formation similar to *sgl1* flowers [47,77]. On the other hand, AtSTM was shown recently to function together in *A. thaliana* with AP1 to specify floral meristem identity in part via UFO [78]. This suggests that SGL1 and STM pathways may converge on MtUFO and that a competition for UFO interaction or for targets shared between SGL1 and AtSTM could be the basis for the higher effect of AtSTM in the absence of SGL1.

Our work shows that AtSTM substitutes for SGL1 function *in M. truncatula* during both vegetative and reproductive development. A parallel has been proposed between compound leaflet primordia and common primordia formation. Both of these processes seem to require the maintenance of an indeterminate phase controlled by SGL1 [61]. While in leaves, SGL1 maintains the indeterminate state, in flowers, SGL1 acts in opposite by promoting the formation of common primordia. The capacity for AtSTM to substitute for SGL1 in both leaves and flowers underlines this parallel and the control of meristematic activity shared by these two transcription factors.

## 4. Materials and Methods

### 4.1. Plant Growth and Plant Material

*M. truncatula* plants were grown in a greenhouse or in growth chambers under long-day conditions (16 h light at 23 °C and 8 h dark at 15 °C). The wild-type (R108) and the *sgl1-1* mutant *M. truncatula* lines have been described [42].

The *pSGL1:GUS* reporter construct was generated as follows. A 2.7 kb fragment corresponding to the *SGL1* (Medtr3g098560) promoter sequence used in [42] (wild-type *M. truncatula* cv Jemalong) was amplified from the *M. truncatula* R108 ecotype using primers pSGL1-for, incorporating a *Bgl*II site, and pSGL1-rev, incorporating a *Bam*HI site. The promoter was cloned into *pCR Blunt II-TOPO* vector to create *pTOPO-pSGL1* and sequenced. The *pSGL1* promoter was moved into the binary vector *pCAMBIA 3301* in front of the *β-glucuronidase* (*GUS*) gene. For this, a *Bgl*II-*Bam*H1 fragment containing the *SGL1* promoter was ligated into *pCAMBIA3301* cut with *Bam*HI and *Bgl*II to replace the *35S* promoter.

The *pSGL1:HA-AtSTM* construct was generated as follows (*AtSTM*, AT1G62360)*. pTOPOpSGL1* was cut with *Eco*R1-*Bam*H1 to release the *pSGL1* promoter, which was cloned into the *pCAMBIA 3300* binary vector cut with *Eco*RI and *BamHI* to create *pCAMBIA 3300 pSGL1*. The *alli2AtSTM* plasmid harboring the triple hemagglutinin (HA) tag-AtSTM fusion under the double enhanced cauliflower Mosaic Virus 35S promoter was used as a template to amplify the HA-AtSTM fusion using primers AtSTM-for and AtSTM-rev incorporating *Bam*HI and *Eco*RI sites, respectively. This fragment was ligated into the pALC vector (Syngenta Ltd., Jeolotts Hill, UK) cut with *Bam*HI and *Eco*RI. The *Bam*HI*-Xba*I fragment containing the HA-AtSTM fusion and the *35S* terminator was cloned into *pCAMBIA 3300 pSGL1* to create *pCAMBIA pSGL1:HA-AtSTM 35S term*. 

The *p35S:HA-AtSTM* construct was generated as follows. The *pSGL1* promoter sequence of the *pCAMBIA pSGL1:HA-AtSTM 35S* term was replaced with the *35S* promoter sequence from *pCAMBIA 3301* using the *Bgl*II and *Bam*H1 sites. The *pCAMBIA 3301* was cut with *Bam*H1 and *Bgl*II to release the 35S promoter, and the *pCAMBIA 3300* containing the *pSGL1:HA-AtSTM* construct was cut with *Bgl*II and *Bam*HI to replace the pSGL1 promoter with the 35S promoter to create *pCAMBIA p35S:HA-AtSTM* 35S term. *pSGL1-GUS*, *p35S:AtSTM* and *pSGL1:AtSTM* constructs were introduced into *A. tumefaciens* GV3101. The *pSGL1-GUS* construct was used to transform *M. truncatula* R108 wild-type plant, while *p35S:AtSTM* and *pSGL1:AtSTM* constructs were used to transform *M. truncatula* R108 plants heterozygous for the *sgl1-1* mutation. *M. truncatula* transgenic lines were created using a leaf disc protocol [79]. Transgenic calli were selected on media containing 3 mgL^−1^ Basta (glufosinate–ammonium). Primers are listed in Appendix A.

Four independent *pSGL1:GUS* transgenic lines were analyzed for SGL1:GUS activity. The SGL1:GUS activity was detected in meristem, vascular tissue and young leaves in R108 *M. truncatula* (Appendix A), which was similar to the activity of the *SGL1* promoter isolated from the JemalongA17 ecotype [42], and in axillary meristem, young floral buds and carpels (Appendix A).

Most of the transgenic plantlets expressing *AtSTM* were not viable when transferred to soil. RT-PCR were realized to compare the level of expression of *AtSTM* in transgenic lines. Total RNA was extracted from *AtSTM* transgenic lines expressing *p35S:AtSTM* (in vitro seedlings and transgenic plants grown in the greenhouse) using Tri reagent (Sigma-Aldrich, Saint-Quentin-Fallavier, France) and treated with DNAse I (Invitrogen, Waltham, MA, USA) according to the manufacturer’s instructions. *AtSTM* levels were monitored using qAtSTM-F and qAtSTM-R primers. Primers specific for the *M. truncatula UBIQUITIN* gene (Medtr3g091400) were used as an internal control [80]. Only transgenic plantlets expressing *AtSTM* at low levels were viable in the greenhouse. Four *p35S:AtSTM* independent lines and three *pSGL1:AtSTM* lines were obtained. Of these, two independent *p35S:AtSTM* lines and one *pSGL1:AtSTM* based on their phenotype were chosen for further characterization. These plants showed reduced fertility. Plants homozygous for the *p35S:AtSTM* construct and heterozygous for *sgl1* were obtained and confirmed by PCR genotyping [42].

### 4.2. Phenotypic Observations

Leaves and flowers were observed under a binocular microscope (Nikon, SMZ1000) and imaged with a digital camera (ProgRes C10^plus^). *M. truncatula* meristems showing GUS activity were dissected and photographed using a LeicaMZ12 dissecting microscope fitted with an AxioCam ICc5 digital camera.

### 4.3. Quantitative Analyses of Leaf Development

Progenies of *SGL1*+/*sgl1* (R108) and *p35S:AtSTM SGL1+/sgl1* lines were grown in a greenhouse. Four-week-old plants were used. The leaflet number was determined on R108 wild-type, *sgl1/sgl1, p35S:AtSTM SGL1+* and *p35S:AtSTM sgl1/sgl1* plants. Eighteen plants per genotype were analyzed.

### 4.4. Scanning Electron Microscopy (SEM)

Three to eight-week-old plants were dissected to observe leaf and flower primordia. The samples were imaged using SEC DESKTOP SEM (Scanning Electron Microscope, (SNE-1500M), SEC, Suwon, Korea) at an accelerating voltage of 15 kV.

### 4.5. In Situ Localization of GUS Activity and In Situ Hybridization

GUS staining and tissue embedding have been described in [81]. RNA in situ hybridization with digoxigenin-labeled probes was performed as previously described [82]. The RNA antisense and sense probes of *MtAP1* (Medtr8g066260) *MtPIM*, *MtPI* (Medtr3g088615) and *MtAGb* (Medtr8g087860) were generated using as cDNA templates a 426 bp fragment of *MtPIM* (282–707 from ATG), a 298 bp fragment of *MtPI* (504–801 from ATG) or a 215 bp fragment of *MtAGb* (558–773 from ATG), respectively, cloned into the pGEM-T Easy vector (Promega, Madison, WI, USA) and using the corresponding SP6 and T7 RNA polymerases in the vector for transcription. SP6 was used for transcription of RNA antisense probes and T7 for the sense. The in situ hybridization with control sense probes is presented in Appendix A.

## Figures and Tables

**Figure 1 ijms-23-14114-f001:**
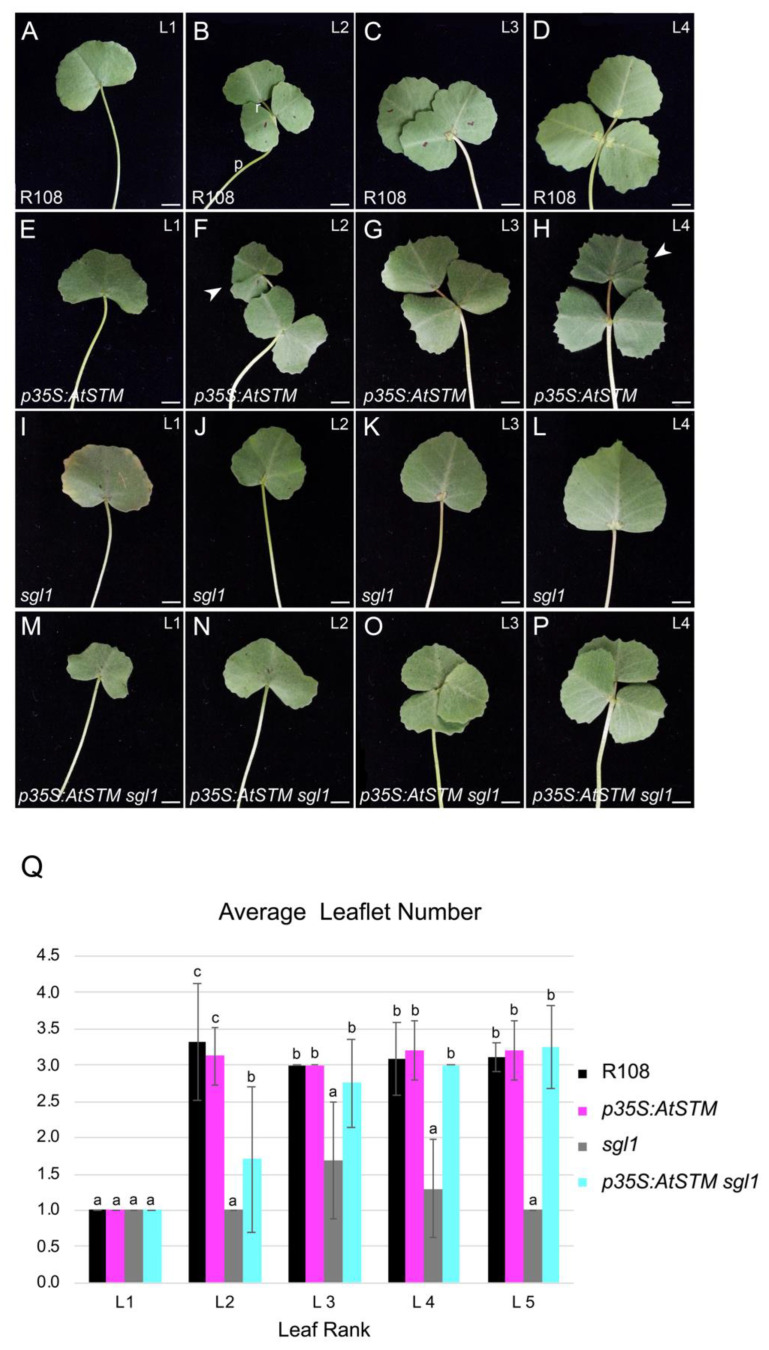
Ectopic expression of *AtSTM* rescues the *sgl1* leaf phenotype. Phenotype of juvenile (L1) and adult leaves (rank L2, L3, L4) of 5-week-old plants. (**A**–**D**) R108 control line. Juvenile leaves are simple, while adult leaves are trifoliate and composed of a terminal leaflet plus two lateral leaflets. The petiole (p) and the rachis (r) are indicated. (**E**–**H**) *p35S:AtSTM*, a transgenic line expressing *AtSTM* under the *p35S* promoter producing a L2 and a L4 heart-shaped adult leaves with an ectopic leaflet fused to the terminal leaflet (arrowheads). This phenotype was occasionally observed. Leaflet margins are serrated. (**I**–**L**) *sgl1* line, showing simple juvenile (L1) and adult leaves (L2–L4). (**M**–**P**) *p35S:AtSTM sgl1* line, showing trifoliate L3 and L4 leaves similar to wild-type. (**Q**) Quantification of the leaflet number. Four-week-old plants were analyzed (n = 18 plants per genotype). Average ± SD are shown. Lowercase letters indicate significant differences between genotypes at each leaf rank (one-way ANOVA with Tukey’s post hoc test; *p* ≤ 0.001). Bars = 5 mm.

**Figure 2 ijms-23-14114-f002:**
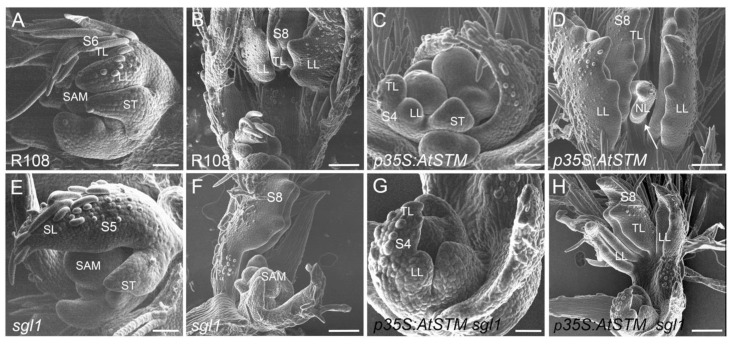
SEM analysis of early stages of leaf development. (**A**,**B**) R108 wild-type control line. (**A**) SAM and a leaf primordia at S6 stage showing a terminal leaflet (TL) developing trichomes with one of the lateral leaflets (LL) and one stipule (ST). (**B**) S8 stage leaf primordia with one terminal leaflet folded on itself between the two lateral leaflets. (**C**,**D**) *p35S:AtSTM* line. (**C**) SAM with a typical S4 stage leaf primordia. (**D**) At S8, the *p35S:AtSTM* line has formed a new leaflet (NL) at the base of the terminal leaflet (arrow). Leaflet margins are dissected. (**E**,**F**) *sgl1* mutant. (**E**) SAM with a simple S5 leaf primordia (SL). (**F**) At S8, the leaflet is folded on itself. (**G**,**H**) *p35S:AtSTM sgl1* line. (**G**) SAM with S4 leaf primordia that has formed lateral leaflets similar to wild-type. (**H**) At S8, the *p35S:AtSTM sgl1* leaf primordia was similar to wild-type with the terminal leaflet surrounded by two lateral leaflets. Bars = 100 µm.

**Figure 3 ijms-23-14114-f003:**
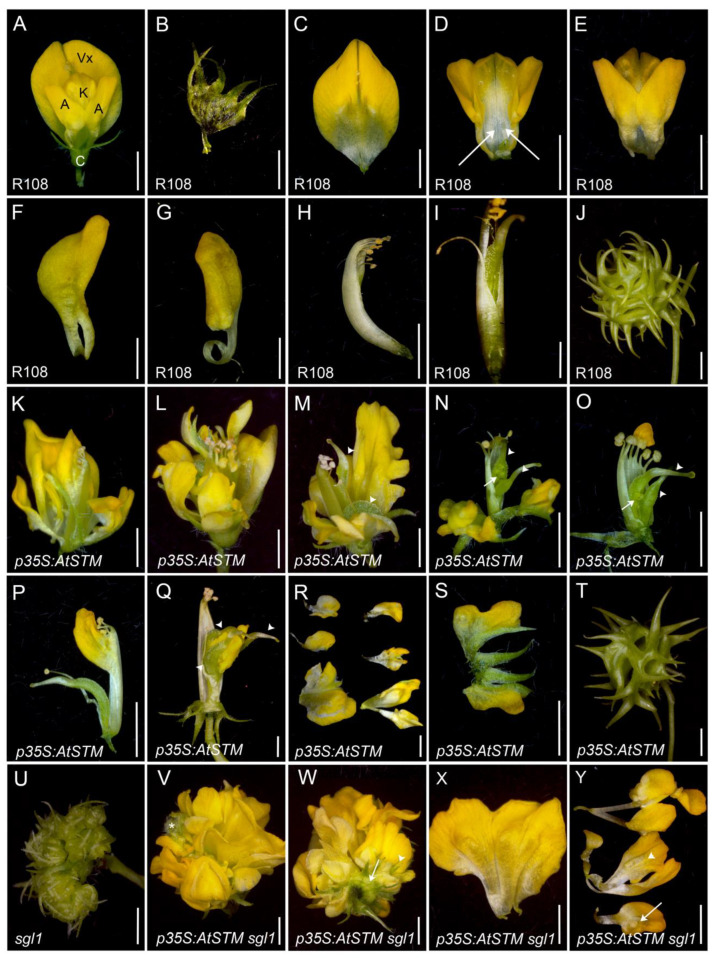
Ectopic expression of *AtSTM* in *M. truncatula* promotes petal identity. (**A**–**J**) R108 wild-type control line. (**A**) The wild-type flower showing the calyx (C) and the corolla containing 5 petals: the standard or vexillium (Vx), the keel (K) and two alae (A) or wings. (**B**–**I**) A wild-type dissected flower: (**B**) the calyx, formed by 5 fused sepals at their base. (**C**) The standard or vexillium, abaxial side. (**D**,**E**) The keel formed by two fused petals (arrows) surrounded by two lateral petals, the alae or wings, adaxial (**D**) and abaxial (**E**) sides, (**F**) a dissected wing. (**G**) A dissected keel petal. (**H**,**I**) A single carpel enclosed by a staminal tube comprising nine fused stamens plus one independent “vexillary” stamen at the adaxial position (**I**). (**J**) After fertilization, the carpel grows out to form a coiled fruit with spines. (**K**–**T**) p*35S:AtSTM* line. (**K**–**Q**) Phenotypes of *p35S:AtSTM* flowers. (**L,M**) Petals were abnormal and can show dissected margins. (**N**) Flower showing petaloid sepals. (**M**–**O**,**Q**) Flowers forming two or three carpels (arrowheads), with some showing an unfused carpel (arrows). (**O**,**P**) Flowers showing petaloid stamens. (**Q**) A dissected flower (the corolla was removed) showing 3 carpels with one developing petaloid sectors. (**R**) Dissected petals, some of them showing sepal sectors. (**S**) A dissected calyx showing petaloid sepals. (**T**) Fruits were smaller with unbent spines. (**U**) *sgl1* inflorescence containing three flowers with a cauliflower-like morphology, *sgl1* flowers contain sepal and carpeloid structures and lack petals and stamens. (**V**–**Y**) A *p35S:AtSTM sgl1* flower showing petals. This flower contains inside incomplete FMs that produce mainly petals or petaloid sepals and a few sepals. (**V**) FMs are visible (*). (**W**) Bottom view showing the calyx (arrow), the sepal form is restored (see also Figure 4M), some other sepals are visible (arrowhead). Dissected petals: a vexillium-like petal (**X**), a keel-like (arrow Y) and wing-like petals (arrowhead Y). Bars = 2 mm, except for F, G, I, N and U, for which bars = 1 mm.

**Figure 4 ijms-23-14114-f004:**
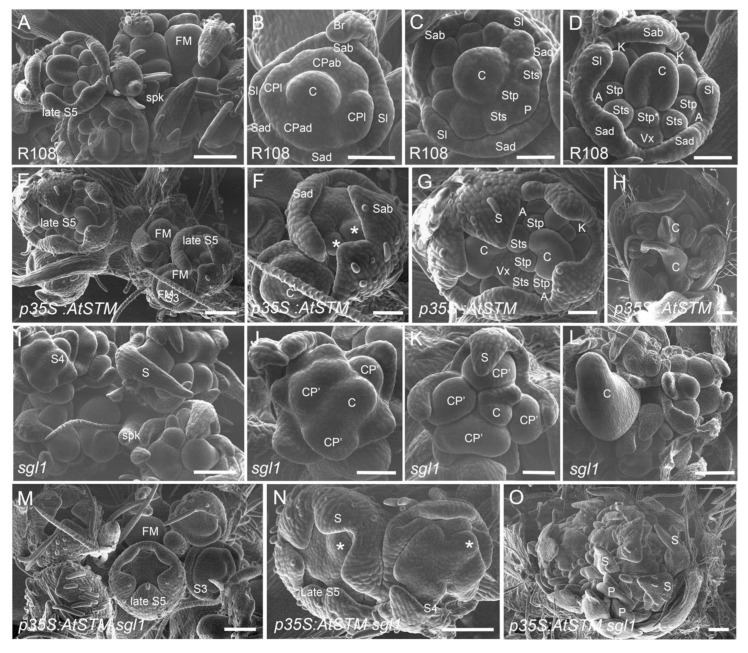
SEM analysis of flower development. (**A**–**D**) Flower development in wild-type R108 line. (**A**) An inflorescence showing floral meristems (FMs) at different stages, including a late S5 stage primordia and a spike (spk) at the base of the floral meristem. (**B**) S4 stage FM showing the abaxial sepal (Sab), the adaxial sepal (Sad), two lateral sepals (Sl), four common primordia (CPab, CPad and two CPI) and one carpel primordia (Cp), Br (bract). (**C**) S5 stage FM showing antesepal stamen (Sts), antepetal stamen (Stp) and petal (P) primordia. (**D**) A late S5 stage FM showing differentiated flower organ primordia, non-fused vexillary stamen (stp*), alae petals (A), vexillum (Vx), keel petals (K), carpel (C). (**E**–**H**) Flower development in *p35S:AtSTM* plants. (**E**) Inflorescence showing floral meristems at different stages. (**F**) Close-up of a late S5 stage FM showing the differentiation of sepal primordia (S) and the formation of bulges in the center (*). Other floral organ primordia are not differentiated. (**G**) Close-up of a late S5 FM showing two carpel primordia (C). (**H**) S8 stage flower with two differentiated carpels (C). (**I**–**L**) Flower development in *sgl1* plants. (**I**) *sgl1* inflorescence showing multiple incomplete FMs, elongated sepals (S) and S4 stage FM. (**J**) A close-up view of S4 stage *sgl1* FM showing defective common primordia (CP’). (**K**) S5 stage FM, CP’s do not further differentiate. (**L**) Carpel primordium (C). (**M**–**O**) Flower development in *p35S:AtSTM sgl1* plants. (**M**) Inflorescence showing incomplete FMs at different stages. (**N**) Close-up of S4 and a late S5 stage incomplete FM showing the differentiated sepals and bulges in the center (*). Note that the sepal form is similar to wild-type sepal (see (**D**)). (**O**) S8 stage flower (the calyx has been removed) containing two FMs that differentiate sepals (S) and petals (P). An elongated sepal is visible. Bars: (**A**,**E**,**I**,**M**–**O**) = 100 µm, (**B**–**D**,**E**,**G**,**J**,**K**) = 50 µm, (**L**) = 250 µm.

**Figure 5 ijms-23-14114-f005:**
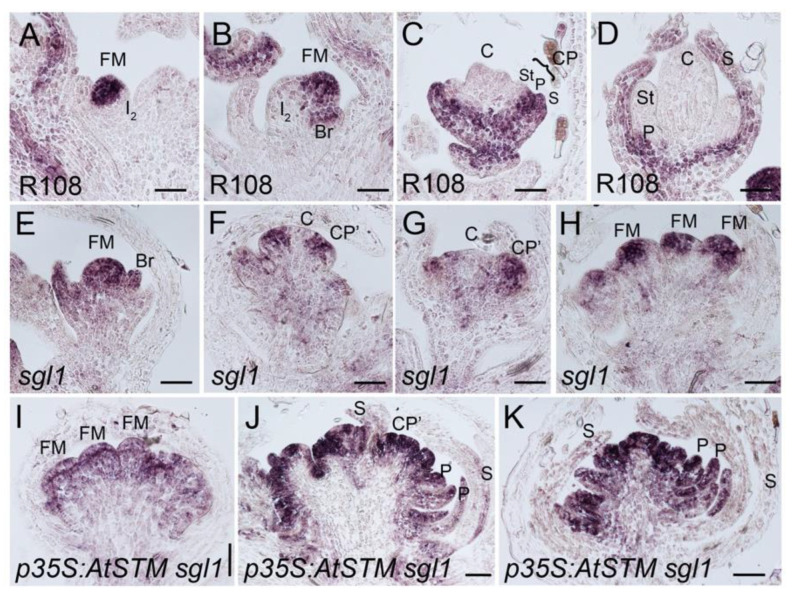
*MtAP1* expression in wild-type, *sgl1* and *p35S:AtSTM sgl1* flowers. (**A**–**D**) R108 wild-type flowers. (**A**,**B**) *MtAP1* was expressed in the floral meristem (FM) and in the bract primordia (Br) and absent in secondary inflorescence meristem (I2). (**C**) At stage 4, *MtAP1* was expressed in sepals (S). *MtAP1* was restricted to the outer part of the developing common primordia (CP), which will give rise to petal (P) and was absent in the inner part that will give rise to stamen (St). (**D**) At stage 6, *MtAP1* expression was maintained in sepals and petals. *MtAP1* was absent in carpels (**C**) and stamens. (**E**–**H**) *sgl1* flowers. (**E**) *MtAP1* was expressed in bract (Br) and floral meristem (FM). (**F**,**G**) *MtAP1* was uniformly expressed in defective common primordia (CP’), unlike in wild-type common primordia, *MtAP1* was absent in the inner part of the floral meristem where carpel will develop (C). (**H**) *MtAP1* was expressed in reiterated floral meristems deriving from floral primordia. (**I**–**K**) *p35S:AtSTM sgl1* flowers. *MtAP1* expression was detected in reiterated floral meristems (FM), in defective common primordia (CP’) and in developing petals (P). Bars: (**A**–**I**) = 50 µm, (**J**,**K**) = 100 µm.

**Figure 6 ijms-23-14114-f006:**
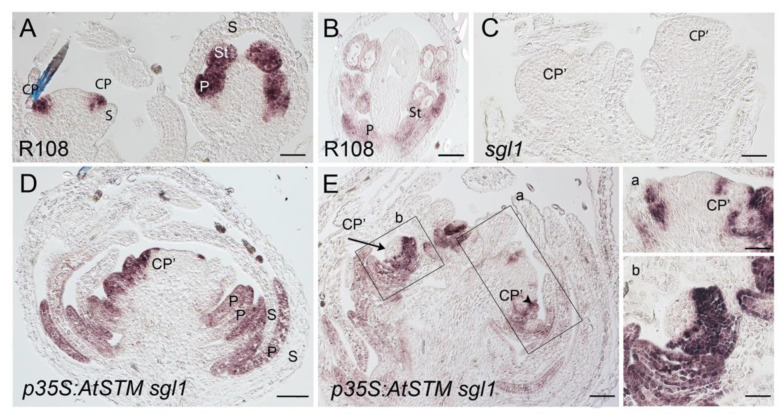
*MtPI* expression in wild-type, *sgl1* and *p35S:AtSTM sgl1* flowers. (**A**,**B**) *MtPI* expression in R108 wild-type flowers. (**A**) *Mt PI* expression was detected at stage 3 in cells of the common primordia (CP). (**B**) At later stages, *MtPI* expression was restricted to stamens (St) and petals (P). (**C**) *MtPI* was not detected in *sgl1* flowers. (**D**,**E**) *MtPI* expression in *p35S:AtSTM sgl1* flowers. (**D**) A flower developing sepals (S) and petals (P). *MtPI* expression was detected in defective common primordia (CP’) and in petals (P). (**E**) A cauliflower showing several floral meristems and developing sepals (S) and petals (P). *MtPI* was detected in defective common primordia (CP’) and in petals (P). Unlike in wild-type, in which the expression of *MtPI* is observed in the whole common primordia, the expression of *MtPI* was restricted to the periphery of the defective common primordia, which will give rise to petals (see details in (**a**,**b**)). (**a,b**) Close-ups of the areas marked in (**E**). Bars: (**A**,**C**,**a**,**b**) = 50 µm, (**B**–**E**) = 100 µM.

**Figure 7 ijms-23-14114-f007:**
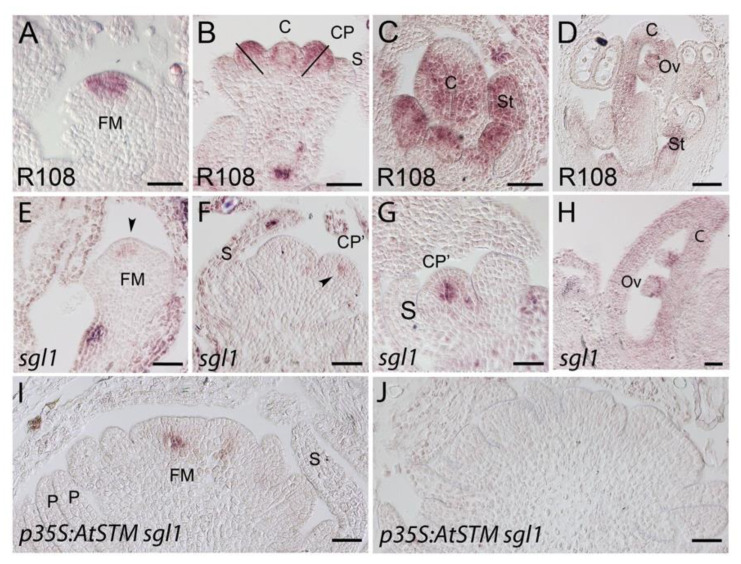
*MtAGb* expression in wild-type, *sgl1* and *p35S:AtSTM sgl1* flowers. (**A**–**D**) *MtAGb* expression in wild-type flowers. (**A**) *MtAGb* expression was located in the center cells of the floral meristem (FM) at stage 2. (**B**) At stage 4, *MtAGb* expression was detected in the carpel primordia (**C**) and the half of the common primordia (CP) that will give rise to the stamens. (**C**) At stage 5, *MtAGb* expression was detected in stamen (St) and carpel (**C**) primordia. (**D**) In later stages, *MtAgb* expression was located in stamen (St), carpel (**C**) and developing ovules (Ov). (**E**–**H**) *MtAGb* expression in *sgl1* flowers. *MtAGb* was detected in FM (**E**, arrowhead) and in defective common primordia (CP’, **F** arrowhead, **G**) and absent in the L1 layer. (**H**) In later stages, *MtAGb* expression was detected in carpel (**C**) and ovules (Ov). (**I**,**J**) *MtAGb* expression in *p35S:AtSTM sgl1* flowers. Expression was detected in 3 flowers out of 13. (**I**) An apex showing expression underneath the outermost cell layers in floral meristem (FM). (**J**) An apex showing no expression. Bars = 50 µm.

## Data Availability

Not applicable.

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
