# Peer review of "Arabidopsis thaliana SHOOT MERISTEMLESS Substitutes for Medicago truncatula SINGLE LEAFLET1 to Form Complex Leaves and Petals"

_ijms, 2022, doi:10.3390/ijms232214114_

Round 1
Reviewer 1 Report
In this article, the authors compared the activity of two transcription factors AtSTM and SGL1 in M.truncatula and found that AtSTM could substitute for SGL1 function in M. truncatula during both vegetative and reproductive development. Their work reveals a function for AtSTM in floral organ identity and there may be have a higher level of interchangeability between meristematic and floral identity functions for the AtSTM and SGL1 transcription factors than previously thought. Overall, their results are interesting and suitable for publication in this journal. In the meantime, there are a few aspects that need to be addressed in the revision.
1.The authors have compared the expression of several floral organ identity among wild type, sgl1 and p35S:AtSTM sgl1, I want to know whether the authors can also provide information about the expression of critical genes that control leaf complexity and then reveal how AtSTM substitutes for SGL1 in the formation of compound leaves.
2. Could the authors provides some information about the direct interaction between AtSTM or SGL1 and their target genes? Whether can they target the same genes that control leaf development or floral meristem identity?
Author Response
Rev1
In this article, the authors compared the activity of two transcription factors AtSTM and SGL1 in M.truncatula and found that AtSTM could substitute for SGL1 function in M. truncatula during both vegetative and reproductive development. Their work reveals a function for AtSTM in floral organ identity and there may be have a higher level of interchangeability between meristematic and floral identity functions for the AtSTM and SGL1 transcription factors than previously thought. Overall, their results are interesting and suitable for publication in this journal. In the meantime, there are a few aspects that need to be addressed in the revision.
1.The authors have compared the expression of several floral organ identity among wild type, sgl1 and p35S:AtSTM sgl1, I want to know whether the authors can also provide information about the expression of critical genes that control leaf complexity and then reveal how AtSTM substitutes for SGL1 in the formation of compound leaves.
We did not monitor the expression of genes that control leaf complexity in p35S:AtSTM sgl1, sgl1 and in wild type leaves as it was previously shown that the ectopic expression of the tomato LeT6 KNOXgene into Medicago sativa increases leaf complexity (Champagne et al. 2007). In addition, Zhou et al. 2014 investigated the interaction between MtKNOX members and ASYMMETRIC LEAVES1/ROUGHSHEATH2/PHANTASTICA and showed that MtKNOX members in Medicago truncatula increases leaf complexity (Zhou et al. 2014). For this reason, we rather decided to focus our characterization of the floral phenotype induced by AtSTM, which was not previously described.
Understanding how AtSTM can substitute for SGL1 for the formation of compound leaves would require doing a comparative transcriptomic analysis which is beyond the scope of this manuscript.
2. Could the authors provides some information about the direct interaction between AtSTM or SGL1 and their target genes? Whether can they target the same genes that control leaf development or floral meristem identity?
Thank you for this interesting point, but we don't know about the direct interactions. It is likely that different target genes are involved in leaf and floral meristem identity. As mentioned above, this would require a large-scale transcriptomic study which is beyond the scope of this paper.
In leaves, LEAFY/SGL1 and KNOX can similarly increase leaf complexity (through the formation of leaflets or serrations). Some species are able to respond to both factors. Similar to its role in the SAM during leaf primordia initiation, the formation of serrations depends on auxin and involves the KNOX1-PIN-FORMED1 (PIN1)-CUC2 module (Kawamura et al., 2010; Bilsborough et al., 2011). Variations in the activity of this module shape the leaf (Maugarny-Cales and Laufs,163 2018). CUC2 is also a direct target of LFY in flower meristems (Winter et al. 2011) (Wu et al. 2012) (Yamagushi et al. 2014).
In flowers, we do not know whether the interaction between AtSTM or SGL1with AP1 or PISTILLATA is direct. In a recent study, Cheng et al. 2018 described the synergistic interaction of SGL1 with AP1 in Medicago truncatula and showed that AP1 expression is reduced in sgl1 mutant. Micro array results indicate that class B genes were dramatically downregulated in sgl1. This indicates that SGL1 is required for petal/stamen specification from common primordia by repressing MtAP1 expression and upregulating class B function genes Our data showing that STM substitutes for SGL1 in promoting AP1 and B function are consistent with Chen et al data.
Reviewer 2 Report
The article by Pautot et al., provides show that gene from A. thaliana can replace the function of M. trancatula in the context of leaf complexity. The authors have done an excellent work and the results are clear and logical. Therefore, i recommend this article for publications at IJMS.
My only concern is if the genes are identical then such a compensation of function is expected. Therefore, the authors may think of including sequence-based alignment of these genes and also comment on the possibility of extending the studies to other members of dicots.
Author Response
rev2
The article by Pautot et al., provides show that gene from A. thaliana can replace the function of M. trancatula in the context of leaf complexity. The authors have done an excellent work and the results are clear and logical. Therefore, i recommend this article for publications at IJMS.
My only concern is if the genes are identical then such a compensation of function is expected. Therefore, the authors may think of including sequence-based alignment of these genes and also comment on the possibility of extending the studies to other members of dicots.
STM and SGL1 belong to two distinct transcription factor families with no homology.
STM is a member of the three-amino-acid-loop extension (TALE) class of homeodomain transcription factors. This family contains KNOTTED-like (KNOX) and BEL1-like (BLH or BELL) homeodomain transcription factors, which function as heterodimers. STM is a KNOTTED class I one member.
LEAFY/SGL1 is a plant-specific transcription factor, which is found as a single gene in most land plant species.
Therefore, if STM compensates for SGL1 function in Medicago truncatula, it is likely that STM and SGL1 shared targets.
Reviewer 3 Report
The manuscript presented by Pautot et al. comprehensively demonstrates that LEAFY (SGL1) and KNOX1 (STM from Arabidopsis thaliana) similarly increase leaf complexity in Medicago truncatula and they hypothesize that this is likely due to both genes participating in two distinct pathways that converge on CUC genes. Furthermore, they demonstrate for the first time that these functions shared by LEAFY and KNOX1 also extend to petal formation in M. truncatula. However, there are some aspects of this work that the authors need to clarify, as well as open questions for the future:
- The authors should test whether CUC1/2 (MtNAM) expression levels differ between wild-type and sgl1 mutant plants with and without ectopic expression of AtSTM, as it is supposed to be the integrator of the LEAFY and KNOX1 pathways that increase leaf complexity. They should also perform a similar comparative study with the major floral identity genes.
- Line 175: “In the sgl1 mutant, the majority of leaves are simple”. Could the authors give an estimation of the percentage of compound leaves in the sgl1 mutant line?
- Lines 389-390: “Our analysis is based on transgenic plants that were able to grow in green-house and therefore expressed AtSTM at low levels”. I assume that here the authors are alluding to the results shown in Figure S2D. It would be better to explain that transgenic lines expressing high levels of AtSTM presented a severe phenotype and were not viable in the greenhouse, so only lines with low levels of AtSTM expression could be investigated.
- Lines 413-415: “The effect of AtSTM on petal identity was moderately visible in a SGL1 wild-type background as only few chimeric petaloid floral organs were formed but was spectacular in a sgl1 mutant background”. Although I have no doubt that the results in the sgl1 mutant background were "spectacular", I think that this should be stated in more formal language in the manuscript.
- It would be interesting if, in a future work, they could demonstrate whether ectopic expression of MtSGL1 can increase A. thaliana leaf complexity.
